# Multidimensional predictors of common mental disorders among Indian mothers of 6- to 24-month-old children living in disadvantaged rural villages with women's self-help groups: A cross-sectional analysis

**Samuel Scott**[1,2]*, **Alejandra Arrieta**[1], **Neha Kumar**[1], **Purnima Menon**[1,2], **Agnes Quisumbing**[1]

**1** Poverty, Health, and Nutrition Division, International Food Policy Research Institute, Washington, DC, United States of America, **2** Poverty, Health, and Nutrition Division, International Food Policy Research Institute, New Delhi, India

* samuel.scott@cgiar.org

## Abstract

Common mental disorders (CMD) among mothers cause disability, negatively affect child development, and have high long-term economic costs. Little is known about how factors across multiple life dimensions, modeled together, are differentially related to maternal mental health in high poverty contexts. Further, there is limited evidence on determinants of CMD in areas where self-help groups (SHGs) exist to promote women's wellbeing. Filling this evidence gap is important given the high prevalence of CMD and the rapid expansion of SHGs in rural India. Cross-sectional data were collected from 1644 mother-infant pairs living in disadvantaged rural villages across five Indian states—Jharkhand, Madhya Pradesh, West Bengal, Odisha, and Chhattisgarh—surveyed in the Women Improving Nutrition through Group-based Strategies study. CMD were assessed using the 20-item Self Reporting Questionnaire (SRQ). We examined 31 factors across four life dimensions: work (work type, time spent in labor, domestic and caretaking activities), agency (SHG membership, decision-making, gender attitudes), health/nutrition (underweight, fertility, diet diversity, child illness), and household/environment (dependency ratio, wealth, food security, shocks, water, sanitation). Survey-adjusted multivariate logistic and ordinary least squares regression models were fit to examine predictors of CMD or SRQ score. On average, mothers were 26 (range 18–46) years old and their children were 15 (range 6–24) months old. CMD defined as ≥ 8 positive SRQ responses were reported by 262 women (16%). Protective factors included being engaged in agricultural labor as a main occupation relative to being a housewife (AOR 0.18, 95% CI 0.10–0.32), more time working (0.85, 0.77–0.93), higher decision-making (0.33, 0.16–0.69), SHG membership (0.73, 0.56–0.96), and having an improved toilet (0.49, 0.33–0.72). Risk factors included food insecurity (1.13, 1.07–1.20) and shocks to non-farm livelihoods (2.04, 1.10–3.78). Practitioners and policymakers should aim to improve food security, economic wellbeing and social capital, such as that created through SHG membership, to improve maternal mental health. Future research should aim

**Data Availability Statement:** All relevant data are within the manuscript and its Supporting Information files.

**Funding:** This work was supported by the Bill & Melinda Gates Foundation, Grant Number: OPP1132181 and the CGIAR Research Program on Agriculture for Nutrition and Health.

**Competing interests:** The authors have declared that no competing interests exist.

to understand why working outside the home, albeit in agricultural work, appears to protect maternal mental health in this context.

## Introduction

Common mental disorders (CMD) such as depression and anxiety are important yet often neglected issues in high poverty contexts, where problems that are easier to see, measure, or treat often draw public health investments. Depression is the leading cause of disability worldwide [1] and CMD have been linked to lost productivity [2], increased healthcare expenses [3], poor physical and cognitive development in children [4], and high long-term costs [5]. In India, a systematic review found that 20% of mothers suffer from postpartum depression [6], but little information is available on the prevalence of CMD in mothers beyond the postpartum period. Both biological (epigenetics, fetal growth, microbiome, hormones, neurotransmitters) and psychosocial (mother-child interactions, parental stress) factors mediate the association between maternal CMD and child development [7]. Although similar mechanisms are at play in low- and high-income countries, contextual risk factors for CMD (nutritional deficits, illness, intimate partner violence, poverty, unplanned pregnancy, etc.) are increased in low-income contexts. Further, despite the high burden of disease attributable to poor mental health and wellbeing, programs addressing these issues are extremely under-resourced, particularly in low-income countries [8]. In India, the government's flagship National Programme for Mental Health received 0.07% (approximately USD 50,000) of the 2017–18 health budget [9]. Further, the combination of limited mental health providers, poor access to mental health services, and a culture of taboo about mental health issues in India results in very few cases being identified and treated [10].

Multiple determinants of CMD have been described, including demographic (age, gender, ethnicity), biological (genetic), economic (income, assets, employment), neighborhood (crowding, safety, infrastructure), environmental (trauma, distress, climate, conflict) social and cultural (education, relationships, intimate partner violence, social capital, social networks) factors [11]. This suggests that the risk factor profile, and solutions to address CMD, must be context specific. In the Indian context, psychiatrists have emphasized that, rather than focusing solely on individual "lifestyle" factors, assessment of the risk factors for mental health needs to recognize the broader sociocultural, environmental, and economic factors that affect women living in rural communities [12]. Public programs to address CMD should consider these multiple risk factors to be effective. Further, these factors should be examined simultaneously rather than in isolation to avoid risks of confounding.

The CMD literature accounting for social aspects of women's lives in India shows that participation in community groups may be an effective strategy to prevent and treat CMD in areas with poor access to mental health services [13]. Such groups may be important sources of social support, especially if women marry outside their natal villages, which is common [14]. Across India, women's social lives have the potential to be shaped by self-help group (SHG) meetings, where 10–20 women meet regularly to discuss a range of issues that they face. These issues vary by community, but often relate to women's empowerment, political participation, economic livelihoods, family, community, and health. As of April 2019, the National Rural Livelihoods Mission reported that 5.48 million SHGs were formed [15]. The massive scale of the program has positioned the SHG as a delivery point for at-scale women-focused interventions across sectors including health. Such interventions, in turn, may protect against CMD

through multiple avenues. The current study aims to examine factors across multiple aspects of life associated with CMD among mothers of young children living in disadvantaged areas of rural India where SHGs exist.

## Materials and methods

### Setting and study design

This study uses data from a cross-sectional survey carried out in 158 villages in eight districts across five Indian states: Madhya Pradesh, Jharkhand, Orissa, West Bengal, and Chhattisgarh. The survey was conducted between November 2017 and January 2018 as a sub-study within a larger parent study, Women Improving Nutrition through Group-based Strategies (WINGS). WINGS is a five-year evaluation (2015–2020) of a nutrition behavior change communication intervention co-implemented by Professional Assistance for Development Action (PRADAN) and Public Health Resource Network (PHRN), both non-governmental organizations based in India. PRADAN organizes and strengthens women's self-help groups (SHGs) primarily on agricultural livelihoods and women's wellbeing and PHRN has brought a nutrition focus to these SHGs. The WINGS evaluation follows a quasi-experimental design with blocks (geographic units within districts) assigned to either a nutrition intervention arm or standard arm. Both study arms receive SHG-based livelihood strengthening; the nutrition intervention arm also receives nutrition-related behavior change communication from a trained female volunteer. For the current sub-study, a list of villages where PRADAN operates was obtained and villages were randomly selected from this list. In each randomly selected village, a census of households was conducted and households were invited to participate until the target number of respondents was achieved. The target number for the survey was determined by power calculations done to estimate impact on infant and young child feeding practices. This study includes data from 1644 women between the ages of 18 and 46 years with children aged 6–24 months. The only inclusion criteria for participation were being a mother of a 6- to 24-month-old child.

### Survey procedure and ethical considerations

The field survey was implemented by Oxford Policy Management (OPM), with support and supervision from International Food Policy Research Institute (IFPRI) researchers. Data were collected in three languages using Computer Assisted Personal Interviewing with tablets. Local health enumerators hired by OPM participated in a three-day census training and 15-day survey training with classroom and field components in October-November 2017. The survey team included 42 enumerators, 14 health investigators and 8 supervisors. All enumerators had to pass three written tests, and automatic logic tests programmed into the data collection software were used for error checking during the data collection phase. Supervisors were required to possess a Bachelor's degree at minimum. Participation in the study was voluntary and written consent was obtained prior to interviews. The institutional review board of IFPRI reviewed and approved the study (IRB 00007490) and the study has therefore been performed in accordance with the ethical standards laid down in the 1964 Declaration of Helsinki and its later amendments.

### Outcomes

Symptoms of common mental disorders were measured using the 20-item self-reporting questionnaire (SRQ) [16]. The SRQ-20 was specifically designed for low- and middle-income country settings and includes dichotomous (yes/no) questions pertaining to feelings or

experiences during the month before the interview. In the current study, presence of CMD was defined as 8 or more positive responses out of 20, consistent with previous studies in India [17, 3, 18, 19]. The two outcomes we considered were SRQ score and a dummy variable for CMD.

## Predictors

Detailed indicator definitions are provided in S1 Table. A broad set of indicators were included acrosssix dimensions: women's work, women's agency, women's own health, nutrition, and reproductive history, child age and health, household social status, poverty, and wealth, shocks experienced in the past year, and controls for individual and household demographic characteristics.

Variables were selected based on plausibility of being associated with maternal psychological wellbeing [20, 21].

*Women's work* indicators included primary occupation (agricultural laborer, housewife); number of hours per day spent on work activities; and proportion of time spent on two types of work (caring for the home and family members, and farm/livestock/employed activities).

*Women's agency* indicators included information on a woman's ability to make her own health and nutrition decisions, her attitude toward gender issues, and whether she was an SHG member (yes/no). Decision-making and gender attitudes indices with a range of 0 to 1 were generated using responses to 12 and 6 questions, respectively. Decision-making questions asked the woman who in the household makes decisions about, for example, how much they worked while pregnant or whether the child is offered eggs. Gender attitudes questions asked whether the woman agreed or disagreed with statements such as "A husband should not let his wife work outside home, even if she would like to do it". Higher scores indicate higher ability of the woman to make her own decisions and having more progressive gender attitudes.

*Women's own health, nutrition, and reproductive history* indicators included maternal underweight status, maternal achievement of minimum dietary diversity, early pregnancy (before 18 years of age), failed pregnancy (ever), and whether she was currently pregnant. Maternal weight was measured using Omron-HN-283 scales to the nearest 100 grams and maternal height was measured using Seca-213 stadiometers to the nearest 0.1 cm. Underweight was defined as body mass index below 18.5 $kg/m^2$. Dietary diversity was measured by asking the woman whether she consumed a list of food groups on the previous day, and minimum dietary diversity was defined as consuming foods from at least 5 of 10 groups [22].

*Child age and health* variables included whether the mother had a child aged 6–11 months, and whether the child experienced illnesses in the past two weeks. Child illness in the past two weeks included having at least one of the following symptoms; fever, cough, cold, irregular breathing, or diarrhea.

*Household social status, poverty, and wealth* indicators included caste, wealth, food insecurity, water, and sanitation Caste categories included scheduled caste (SC), scheduled tribe (ST), other backward classes (OBC), or General. Wealth was determined using principal component analysis of household assets, electricity access, and building materials [23]. A dummy variable was created for the poorest 20% of households, the lowest wealth quintile. Food insecurity was measured using the 9-item Household Food Insecurity Access Scale (HFIAS), with categories of mild, moderate and severe food insecurity defined per the HFIAS indicator guide [24]. Water and sanitation indicators were whether the household had an improved drinking water source and improved toilet.

*Shocks* variables include shocks that the household experienced in the past year. Six types of shocks experienced in the past year, were examined: death, illness, demonetization (affected by

government ban of large currency), non-farm livelihoods (loss of employment or business failure), crop loss, and livestock loss.

*Individual and household demographic characteristics* include controls for the woman's age, years of education, household size (number of persons), and dependency ratio. Dependency ratio was the ratio of the number of household members < 16 or >55 years old to the number of household members 16–55 years old. *Interviewer sex* was included to capture its effect on responses to sensitive survey questions, and study arm to control for any influence of the nutrition behavior change communication intervention.

## Statistical analysis

Three methods were used to examine associations between maternal psychological wellbeing and demographic, occupational, social, health, and environmental factors. First, differences between groups with and without CMD were assessed using t-tests. Second, multivariate ordinary least squares regression (OLS) models were fit to examine associations between predictors and total SRQ score. Third, the odds of CMD were predicted using multivariate logistic regression models. All regression models included cluster controls and district fixed effects in addition to the predictors identified above. The analysis also implements F-tests to ascertain whether the coefficients of the groups of related variables discussed above are jointly equal to zero.

Statistical significance was considered at $p < 0.05$ for OLS coefficients or if the 95% confidence interval did not cross 1 for odds ratios. Analyses were conducted in STATA software version 15.1.

## Results

### Sample characteristics

The mean SRQ score was 3.6 out of 20 and CMD were reported by 262 of 1644 women (16%) (Table 1). Women were 25.6 years of age and had 4.9 years of education on average. Most women reported that being a housewife (60%) or farmer/agricultural laborer (30%) was their primary occupation. Of the 10.7 hours of work per day that women reported working, 81% was spent on household chores or caring for family members and 19% on farming, livestock, or employed activities. Women scored 0.7–0.8 out of 1.0 on normalized decision making and gender attitudes indices on average. SHG membership was reported by 43% of the women. Nearly all (98%) women were from disadvantaged social groups, with the majority belonging to tribal groups (ST; 60%) followed by OBC (23%) and SC (14%). Women's nutrition was generally poor, with 39% of women being underweight and 70% not achieving minimum dietary diversity. Early pregnancy was reported by 13% of women and 29% ever had a failed pregnancy. Half (47%) of the children were female and they were 14.9 months of age on average. Child illness was common, with 29% of women reporting that their children had been sick in the previous two weeks. On average, households had 5.5 persons and 50% experienced at least mild food insecurity, though severe food insecurity was uncommon (6%). Two thirds (68%) of households had an improved drinking water source and one third (35%) had an improved toilet. The most common types of shocks experienced by households in the previous year were crop loss (29%), illness of a family member (23%), consequences of demonetization (19%), and livestock loss (18%).

**Table 1. Characteristics of Indian women with children aged 6–24 months for overall sample and by absence and presence of common mental disorder symptoms.**

| | Overall | No CMD (SRQ < 8) | CMD (SRQ ≥ 8) | Group difference |
|---|---|---|---|---|
| | n = 1,644 | n = 1,382 | n = 262 | |
| | *Mean(SD)/%* | *Mean(SD)/%* | *Mean(SD)/%* | *P* |
| **Outcome** | | | | |
| SRQ score | 3.6 (3.8) | 2.3 (2.2) | 10.5 (2.7) | 0.00 |
| **Women's work** | | | | |
| Occupation | | | | |
| Housewife, % | 60 | 57 | 73 | 0.00 |
| Farmer, % | 30 | 33 | 19 | 0.00 |
| Other, %[1] | 10 | 10 | 9 | 0.44 |
| Work time per day, hours | 10.7 (2.3) | 10.8 (2.3) | 10.3 (2.1) | 0.02 |
| HH chores/care, proportion of work time | 81 | 81 | 82 | 0.47 |
| Labor-related work, proportion of work time | 19 | 19 | 18 | 0.47 |
| **Women's agency** | | | | |
| Decision making score, 0–1[2] | 0.8 (0.3) | 0.8 (0.3) | 0.7 (0.3) | 0.02 |
| Progressive gender attitudes score, 0–1[2] | 0.7 (0.2) | 0.7 (0.2) | 0.8 (0.3) | 0.06 |
| Self-help group member, % | 43 | 44 | 38 | 0.04 |
| **Woman's own health, nutrition, and reproductive history** | | | | |
| Woman's nutritional status[3,4] | | | | |
| Underweight, % | 39 | 40 | 38 | 0.71 |
| Normal weight, % | 56 | 56 | 56 | 0.94 |
| Overweight, % | 4 | 4 | 4 | 0.74 |
| Obese, % | 0 | 0 | 1 | 0.24 |
| Woman achieved minimum dietary diversity, % | 30 | 30 | 31 | 0.95 |
| Pregnant before age 18 years, % | 13 | 13 | 13 | 0.97 |
| Ever had failed pregnancy, % | 29 | 27 | 37 | 0.00 |
| Currently pregnant, % | 8 | 8 | 9 | 0.83 |
| **Child age and health** | | | | |
| Child female, % | 47 | 47 | 47 | 0.84 |
| Child age, months | 14.9 (5.0) | 14.9 (5.0) | 14.8 (5.3) | 0.87 |
| Child 6–11 months old, % | 30 | 30 | 33 | 0.26 |
| Child sick in last 2 weeks, %[5] | 29 | 28 | 31 | 0.37 |
| **Household social status, poverty and wealth** | | | | |
| Caste | | | | |
| OBC % | 23 | 23 | 22 | 0.79 |
| General, % | 2 | 3 | 1 | 0.11 |
| SC, % | 14 | 14 | 15 | 0.90 |
| ST, % | 60 | 60 | 62 | 0.72 |
| Poorest wealth quintile, % | 19 | 19 | 20 | 0.77 |
| Food insecurity scale, 0–27[7] | 2.5 (3.4) | 2.3 (3.3) | 3.3 (3.7) | 0.01 |
| Food security[7] | | | | |
| Food secure, % | 50 | 53 | 36 | 0.00 |
| Mild food insecurity, % | 20 | 19 | 21 | 0.51 |
| Moderate food insecurity, % | 23 | 22 | 32 | 0.01 |
| Severe food insecurity, % | 6 | 6 | 11 | 0.09 |
| Water and sanitation | | | | |
| Improved drinking water source, % | 68 | 69 | 60 | 0.07 |
| Improved toilet at household, % | 35 | 37 | 27 | 0.03 |

*(Continued)*

**Table 1.** (Continued)

|  | Overall | No CMD (SRQ < 8) | CMD (SRQ ≥ 8) | Group difference |
|---|---|---|---|---|
|  | n = 1,644 | n = 1,382 | n = 262 |  |
| **Shocks experienced in past year** |  |  |  |  |
| Death, % | 6 | 6 | 5 | 0.42 |
| Illness, % | 23 | 23 | 23 | 0.84 |
| Demonetization, % | 19 | 19 | 16 | 0.45 |
| Non-farm livelihoods, % | 5 | 4 | 8 | 0.01 |
| Crop loss, % | 29 | 30 | 28 | 0.82 |
| Livestock loss, % | 18 | 18 | 17 | 0.74 |
| **Individual and household demographics** |  |  |  |  |
| Woman's age, years | 25.6 (4.5) | 25.5 (4.6) | 25.7 (4.4) | 0.59 |
| Woman's education, years | 4.9 (4.3) | 4.9 (4.4) | 5.0 (4.2) | 0.80 |
| Household size, persons | 5.5 (1.8) | 5.5 (1.8) | 5.7 (1.9) | 0.26 |
| Dependency ratio[6] | 1.1 (0.7) | 1.1 (0.7) | 1.1 (0.7) | 0.92 |
| Female interviewer, % | 97 | 96 | 99 | 0.01 |
| Nutrition intensive arm, % | 49 | 48 | 50 | 0.70 |

[1] Other occupations included: Non-agricultural day laborer (5.9%), service/salaried worker (2.2%), migrant laborer (0.4%), business/traders (0.3%) and small/cottage industry (0.2%).

[2] Decision making score and progressive gender attitudes were based on 12 and 6 questions respectively; both scores were rescaled from 0 to 1.

[3] Standard World Health Organization categorization was used for underweight (BMI<18.5 kg/m$^2$), normal weight (BMI 18.5–24.9 kg/m$^2$), overweight (BMI 25–29.9 kg/m$^2$).

[4] The sample size with anthropometric measures was slightly smaller (n = 1,412 overall; n = 1,206 low CMD; n = 206 high CMD).

[5] Coded as positive if mother reported that child suffered from cough, fever or diarrhea in the previous 2 weeks.

[6] Ratio of individuals aged <15yrs or >55yrs to those aged 16-55yrs.

[7] Measured using the Household Food Insecurity Access Scale (HFIAS), scaled from 0 to 27 with levels of food security defined according to USAID's FANTA III HFIAS guide (2007).

[8] P-values reported are from t-tests for continuous variables, and from chi2 tests for binary variables.

List of abbreviations: CMD, common mental disorders; OBC, other backward class; SC, scheduled caste; ST, scheduled tribe; SRQ, self-reporting questionnaire.

## Multidimensional predictors of common mental disorders

Table 1 presents results from the test of differences between groups who did or did not report CMD symptoms, Table 2 presents results from the OLS regression models used to predict continuous SRQ score, and Fig 1 presents results from the logistic regression models used to predict the odds of reporting CMD symptoms. Findings from the three methods were congruent.

*Women's work*: Joint F-tests of women's work coefficients indicate that women's work variables are jointly significant in explaining women's CMD (p<0.001). Relative to women who reported their main occupation as housewife, those who report mainly being farmers were 82% less likely to report symptoms of CMD (adjusted odds ratio (AOR) 0.18, 95% confidence interval (95% CI) 0.10–0.32) (Fig 1). One additional hour of any type of work per day was associated with a 15% lower likelihood of reporting CMD symptoms (AOR 0.85, 95% CI 0.77–0.93). A non-significant association was observed between spending more work time on household chores and caring for family members—instead of farm and agriculture work—and lower SRQ (β = -1.37, p = 0.07) (Table 2).

*Women's agency*: Women's agency indicators were likewise jointly significant predictors of CMD (p = 0.001). Negative associations were found between CMD symptoms and decision making (AOR 0.33, 95% CI 0.16–0.69) and being an SHG member (AOR 0.73, 95% CI 0.56–0.96). Progressive gender attitudes score was unrelated to SRQ score or CMD symptoms.

**Table 2. Factors predicting SRQ score in Indian women with children aged 6–24 months[1].**

| Women's work | β | (SE) | P |
|---|---|---|---|
| Occupation | | | |
| Housewife | ref | | |
| Farmer | -1.603 | 0.198 | 0.000 |
| Other | -1.024 | 0.352 | 0.011 |
| Work time per day | -0.169 | 0.052 | 0.005 |
| Labor-related work, proportion of work time | ref | | |
| HH chores/care, proportion of work time | -1.371 | 0.708 | 0.072 |
| **Women's agency** | | | |
| Decision making score | -2.048 | 0.487 | 0.001 |
| Progressive gender attitudes score | 0.103 | 0.720 | 0.889 |
| Self-help group member | -0.37 | 0.168 | 0.044 |
| **Woman's own health, nutrition, and reproductive history** | | | |
| Woman underweight | -0.069 | 0.185 | 0.715 |
| Woman achieved minimum dietary diversity | 0.260 | 0.270 | 0.351 |
| Pregnant before age 18 years | -0.423 | 0.302 | 0.182 |
| Ever had failed pregnancy | 0.348 | 0.224 | 0.140 |
| Currently pregnant | 0.108 | 0.594 | 0.858 |
| **Child age and health** | | | |
| Child 6–11 months old | -0.030 | 0.178 | 0.868 |
| Child sick in last 2 weeks | 0.616 | 0.201 | 0.008 |
| **Household social status, poverty and wealth** | | | |
| Caste | | | |
| OBC/General | ref | | |
| SC | 0.926 | 0.305 | 0.008 |
| ST | 0.329 | 0.247 | 0.202 |
| Poorest wealth quintile | 0.081 | 0.190 | 0.677 |
| Food insecurity scale | 0.144 | 0.050 | 0.011 |
| HH experiences any food insecurity | | | |
| Water and sanitation | | | |
| Improved drinking water source | -0.148 | 0.139 | 0.303 |
| Improved toilet at household | -0.310 | 0.194 | 0.132 |
| **Shocks experienced in past year** | | | |
| Death | -0.147 | 0.481 | 0.764 |
| Illness | 0.090 | 0.291 | 0.762 |
| Demonetization | -0.725 | 0.300 | 0.029 |
| Non-farm livelihoods | 0.126 | 0.279 | 0.658 |
| Crop loss | 0.681 | 0.350 | 0.070 |
| Livestock loss | -0.377 | 0.334 | 0.276 |
| **Individual and household demographics** | | | |
| Woman's age | 0.045 | 0.023 | 0.067 |
| Woman's education | 0.036 | 0.030 | 0.250 |
| HH size | -0.096 | 0.064 | 0.158 |
| Dependency ratio | 0.244 | 0.204 | 0.250 |
| Interviewer sex (1 = female) | 2.632 | 0.366 | 0.000 |
| Nutrition intensive arm | 0.382 | 0.140 | 0.015 |
| Constant | 4.192 | 1.772 | 0.032 |
| Joint tests of coefficients | | *F-statistic* | *P* |

(*Continued*)

**Table 2.** (Continued)

| Women's work | β | (SE) | P |
|---|---|---|---|
| Ho: Women's work variables = 0 | | 33.2 | 0.000 |
| Ho: Women's agency variables = 0 | | 10.6 | 0.001 |
| Ho: Women's own health, nutrition, and reproductive history = 0 | | 1.2 | 0.375 |
| Ho: Child's health and age variables = 0 | | 4.7 | 0.025 |
| Ho: Household social status, poverty, and wealth variables = 0 | | 4.8 | 0.006 |
| Ho: Individual and Household demographic variables = 0 | | 2.7 | 0.068 |
| Ho: Shocks variables = 0 | | 3.2 | 0.030 |
| $R^2$ | 0.154 | | |
| Number of observations | 1,360 | | |

[1] Multivariate ordinary least squares model using continuous SRQ score (0–20) as the outcome, with district fixed effects.

List of abbreviations: CMD, common mental disorders; HH, household; OBC, other backward class; SC, scheduled caste; ST, scheduled tribe; SRQ, self-reporting questionnaire.

*Woman's own health, nutrition, and reproductive history*: Maternal health and nutrition was generally unrelated to maternal psychological wellbeing; joint F-tests on this group of variables fail to reject the null hypothesis that the coefficients are jointly equal to zero (p = 0.375). No associations were observed between SRQ or reporting CMD symptoms and women being underweight, having an early pregnancy, being currently pregnant, or achieving minimum dietary diversity. Women with a failed pregnancy reported a higher likelihood of reporting CMD symptoms compared to those who never experienced it (AOR 1.43, 95% 0.95–2.44).

*Child age and health*: In contrast to maternal health and nutrition, we reject the null hypothesis that child age and health variables were jointly insignificant predictors of CMD (p = 0.025). Although the child being aged 6–11 months was not significantly associated with CMD, recent child illness predicted a 0.6-point higher SRQ score (β = 0.62, p = 0.008) but women with recently ill children were not more likely to report CMD symptoms than those whose children had not been sick in the previous two weeks.

*Household social status, poverty, and wealth*: Household socioeconomic characteristics related to caste, poverty, and wealth were jointly significant predictors of maternal CMD (p = 0.006).Being from the SC group relative to OBC/general groups was associated with having a higher SRQ score (β = 0.93, p = 0.008) but this association did not reach significance in the logistic regression model. Living in a food insecure household was associated with increased odds of reporting CMD symptoms (AOR 2.30, 95% CI 1.43–3.69).Lower CMD was associated with having an improved toilet (AOR 0.49, 95% CI 0.33–0.72) Improving drinking water source, and household wealth were not related to SRQ score or CMD.

Shocks: Shocks experienced in the past year were jointly significantly associated with CMD. Experiencing a non-farm livelihood-related shock in the last year was associated with increased odds of reporting CMD symptoms (AOR 2.04, 95% CI 1.10–3.78). Interestingly, experiencing a demonetization-related shock (AOR 0.47, 95% CI 0.25–0.91) was associated with lower odds of reporting CMD symptoms. Women from households that experienced crop loss tended to report a higher SRQ score (β = 0.68, p = 0.07).

Individual and household demographic variables: These variables were not jointly significant in explaining CMD (p = 0.068). Maternal age and education, household size, and household dependency ratio were not significantly associated with SRQ or CMD symptoms.

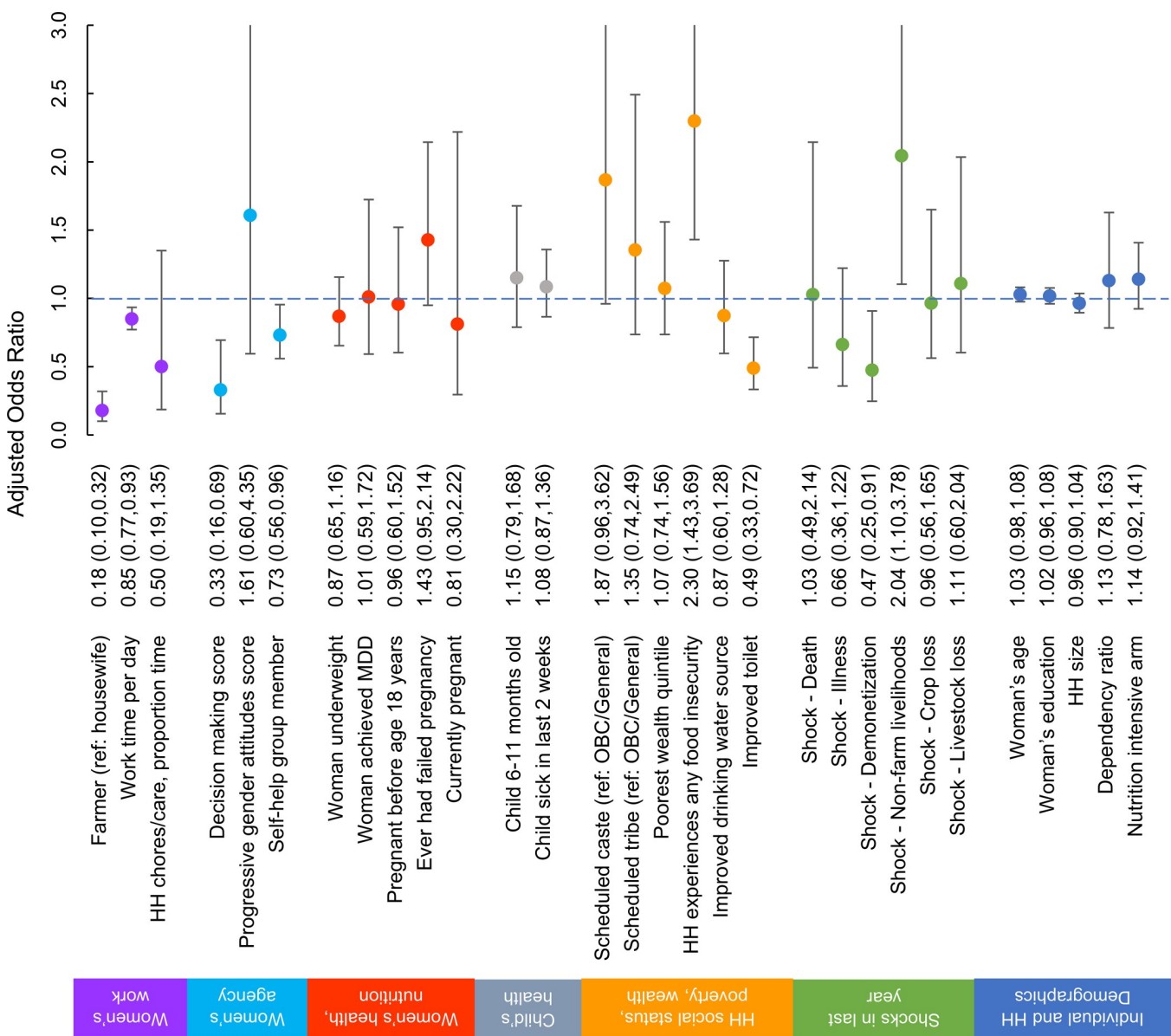

**Fig 1. Factors associated with symptoms of common mental disorders in Indian women with children aged 6–24 months.** Odds ratios were estimated in a multivariate logistic regression and are adjusted for all other factors shown. Factors with confidence intervals which do not cross 1.0 (dotted line) are considered statistically significant. Interviewer sex is omitted from the figure due to not fitting on the same scale, but was significant (AOR 6.22, 95% CI 4.18–9.27). CMD, common mental disorders; HH, household; MDD, minimum dietary diversity; OBC, other backward classes; SHG, self-help group.

*Other factors*: Women who were interviewed by female enumerators were more likely to report CMD symptoms than those interviewed by male enumerators (AOR 6.22, 95% CI 4.18–9.27). Being in the nutrition behavior change communication intervention arm was associated with a slightly higher SRQ score (β = 0.38, p = 0.02) but was not predictive of reporting CMD symptoms.

## Discussion

Our findings highlight that maternal psychological wellbeing in rural India is affected by multiple factors. Risk factors included food insecurity and livelihood shocks. Protective factors

included being engaged in agricultural activities, being able to make one's own decisions, SHG membership, household food security, an improved toilet at home, and not experiencing livelihood shocks. These findings indicate that improvement of women's mental wellbeing will require addressing multiple aspects of their lives.

Differences between farmers and housewives were explored (S2 Table), but it is not clear from the current data why being a farmer for a mother of a young child would be a protective factor for CMD. Plausible explanations may include that spending time outside the home could be a relief from the drudgery of housework or from controlling household members, physical energy spent doing agricultural work in an outdoor environment may relieve stress, or agricultural activities could enhance social connectedness and provide an opportunity for women to have private conversations about personal issues. However, further empirical and ethnographic examination would be useful to better understand the relationship between agricultural labor and mental health.

Women who can make their own choices were less likely to report symptoms of CMD. Low decision-making score may reflect intrahousehold power imbalances where the mother-in-law or the husband have more power relative to the respondent woman. Lack of full scores for decision-making and for progressive gender attitudes suggest the existence of underlying gender inequalities in their living environments. The ability to make choices may also be tied to SHG membership. Participation in SHGs has also been shown to have positive effects on women's economic, social and political empowerment [25]. We are aware of only two studies in India relating SHG participation to mental wellbeing, but neither study used standardized measures of CMD or focused on mothers of young children. In Kerala, a cross sectional study found that SHG membership was predictive of lower stress and higher life satisfaction [26]. In Odisha, a quasi-experimental study using matching methods found no impact of SHG membership on happiness [27]. SHG membership may be related to women's wellbeing through multiple pathways such as boosting income and resources, creating social connections and providing a safe space to discuss issues. Studies reporting impacts of SHG-based interventions on women's mental health [28] should aim to separate the effects of the intervention from effects of group membership.

In our sample, most SHG groups included microcredit in their core activities. In exploratory analyses, we found that women in SHGs with microcredit activities had lower SRQ scores and were less likely to have CMD (S3 Table). We do not have data on women's duration of participation in microcredit activities, but a study with female microcredit participants in Peru found that longer participation was associated with decreased depressive symptoms [29]. A review, albeit from over a decade ago [30], found that supportive networks help mothers acquire appropriate parenting methods, provide tangible resources such as childcare and financial resources, and serve as buffers against stressful life situations. Another study in northern India found that recently delivered women living in villages with SHGs had higher numbers of relationship ties and advice networks [31].

Maternal underweight, early pregnancy, failed pregnancy, and current pregnancy were not found to be associated with CMD symptoms. The SRQ-20 has a recall period of 30 days, thus may not capture the effects of early pregnancy and failed pregnancy on mental health. We did find a marginal effect of higher odds of CMD symptoms for women with a failed pregnancy compared to women without a failed pregnancy. Being underweight is possibly a chronic state that the women may not perceive as noteworthy. At the child level, recent child illness was associated with a higher SRQ score, in line with evidence from Ethiopia, Bangladesh, Vietnam [32, 33] and Pakistan [33]

Food insecurity, being from a backward caste, and lack of improved sanitation were associated with higher SRQ and increased likelihood of CMD. Half of the households in our sample

experienced some level of food insecurity, highlighting the magnitude of this issue in tribal villages of India. The association between food security and women's mental health has been reported in Bangladesh [32, 34], Zambia [35], Vietnam and Ethiopia [32]. Our finding of an association between sanitation and CMD is supported by a study in Odisha which found that women experience multiple sanitation-related stressors that affect their psychosocial wellbeing; greater privacy, lower risk of uro-genital tract infections, and protection from assaults by men were identified as important benefits afforded by having a private sanitation facility [36].

We observed a lower mean SRQ score among women interviewed by men compared to those interviewed by women, suggesting that women may not be as comfortable responding to men. SRQ score was also slightly higher among women in the nutrition intervention arm of the study compared to women in the comparison arm. Odds of reporting CMD symptoms, however, were similar between the two groups. Given that the observational nature of the data, we were not able to ascertain whether the slightly higher SRQ score in the intervention arm was attributable to the intervention.

The primary strengths of the study were the inclusion of an understudied population sampled from multiple states in India and the breadth of measures covering multiple aspects of women's lives in rural high poverty settings. Despite capturing broad measures, we recognize that mental health has a tremendously complex set of determinants which we can only begin to understand in our study. There are several components that we did not capture such as the genetic component of emotional wellbeing, size and quality of social networks, interspousal violence, or earlier life events that may have long-term psychological consequences. Further, we acknowledge that causal inference is not possible given the cross-sectional nature of these data. For example, our study cannot answer the question of whether SHG membership reduces CMD or if barriers to SHG membership are lower in women without CMD. Longitudinal data would be valuable in this regard. Finally, CMD was assessed through self-reported symptoms to an enumerator following an established protocol during a structured, face-to-face interview rather than a through a clinical diagnosis by a mental health professional.

Addressing poor mental health is a worthy public health goal, particularly in mothers who interact with and influence the development of the next generation. Our study adds to the available evidence on the breadth of factors associated with CMD in the context of rural India. To our knowledge, this is the first study to identify an association between SHG membership and CMD in mothers of young children, which is particularly relevant to women in disadvantaged rural areas with limited access to mental health services. As women's and mother's support groups continue to expand in India and globally, the nature of this association should be explored experimentally.

## Supporting information

**S1 Table. Variable definitions.**
(DOCX)

**S2 Table. Characteristics of Indian women with children aged 6–24 months by main occupation.**
(DOCX)

**S3 Table. SRQ scores for SHG members by absence and presence of credit activities in SHG.**
(DOCX)

**S1 File.**
(DTA)

## Acknowledgments

We would like to thank the women in the study for volunteering their time to participate and Oxford Policy Management for coordinating the survey and collecting the data. The authors declare no conflicts of interest. S.S., N.K., P.M., and A.Q. designed the study. S.S. and A.A. conceived the idea for the analysis. A.A. analyzed the data. S.S. prepared the initial version of the manuscript and all authors contributed to the writing and interpretation of results in the final version.

## Author Contributions

**Conceptualization:** Samuel Scott.

**Data curation:** Samuel Scott, Alejandra Arrieta.

**Formal analysis:** Samuel Scott, Alejandra Arrieta, Neha Kumar.

**Funding acquisition:** Neha Kumar, Purnima Menon, Agnes Quisumbing.

**Investigation:** Samuel Scott, Alejandra Arrieta, Neha Kumar.

**Methodology:** Samuel Scott, Alejandra Arrieta, Neha Kumar, Agnes Quisumbing.

**Supervision:** Neha Kumar, Purnima Menon, Agnes Quisumbing.

**Visualization:** Samuel Scott.

**Writing – original draft:** Samuel Scott.

**Writing – review & editing:** Samuel Scott, Alejandra Arrieta, Neha Kumar, Purnima Menon, Agnes Quisumbing.

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
