## [Decision Letter · Decision Letter 0]

9 Jan 2020

PONE-D-19-27808

Multidimensional predictors of common mental disorders among Indian caregivers of 6- to 24-month-old children living in villages with women’s self-help groups

PLOS ONE

Dear Scott,

Thank you for submitting your manuscript to PLOS ONE. After careful consideration, we feel that it has merit but does not fully meet PLOS ONE’s publication criteria as it currently stands. Therefore, we invite you to submit a revised version of the manuscript that addresses the points raised during the review process.

We would appreciate receiving your revised manuscript by Feb 23 2020 11:59PM. To enhance the reproducibility of your results, we recommend that if applicable you deposit your laboratory protocols in protocols.io, where a protocol can be assigned its own identifier (DOI) such that it can be cited independently in the future. For instructions see: http://journals.plos.org/plosone/s/submission-guidelines#loc-laboratory-protocols

We look forward to receiving your revised manuscript.

Kind regards,

Thach Duc Tran, M.Sc., Ph.D.

Academic Editor

PLOS ONE

Journal Requirements:

2. Please provide additional details regarding participant consent. In the ethics statement in the Methods and online submission information, please ensure that you have specified how the verbal consent was documented and witnessed). If your study included minors, state whether you obtained consent from parents or guardians. If the need for consent was waived by the ethics committee, please include this information.

Reviewers' comments:

Reviewer's Responses to Questions

**Comments to the Author**

1. Is the manuscript technically sound, and do the data support the conclusions?

Reviewer #1: Yes

Reviewer #2: Yes

Reviewer #3: Partly

2. Has the statistical analysis been performed appropriately and rigorously? 

Reviewer #1: Yes

Reviewer #2: Yes

Reviewer #3: No

3. Have the authors made all data underlying the findings in their manuscript fully available?

Reviewer #1: Yes

Reviewer #2: Yes

Reviewer #3: Yes

4. Is the manuscript presented in an intelligible fashion and written in standard English?

Reviewer #1: Yes

Reviewer #2: Yes

Reviewer #3: Yes

5. Review Comments to the Author

Reviewer #1: PLOS ONE

Multi-dimensional predictors of common mental disorders among Indian care-givers of 6 to 24-month-old children living in villages with women’s self-help groups.

Thank you very much for giving me an opportunity to review this paper. This study is very important focused on rural area of India and this sample population was quite big. Overall, this paper is well-written and worth publishing.

Here are my comments. Hope these are helpful to improve consistency and persuasiveness.

Title

Please include targeted population (living in disadvantaged area, in rural villages)and study design (cross-sectional study).

The participants included those without self-help groups (Table 1), hence it should be deleted.

Introduction

1. 1st Paragraph: Authors need to describe more about the impact of CMD of care givers in low and middle income countries or Indian cultural contexts. For example, malnourishment of baby or infanticide?

2. 1st Paragraph: In order to address the importance of mental health issued among caregivers, please add the findings of previous studies with regard to the prevalence of CMD or depression among mothers.

3. I wonder if ‘care giver’ includes father, mother and other relatives? If survey assessed mother, please change the term as mother.

4. 2nd Paragraph, please describe more in detail about determinants of CMD in previous studies. Although third paragraph mentioned the literature regarding social factor for women’s lives is scarce, previous studies have examined cultural factors in 2nd paragraph.

5. What is the originality of this study compared with previous studies? Including variable regarding self-help group? What hypothesis can be considered? Attending in self-help group is one of independent factors?

6. Aim: What brought authors to investigate targeted to caregivers of young children living in disadvantages areas of rural area, which reason should be explained in introduction and specified in title and abstract.

Methods

1. Although authors selected disadvantageous areas in aims, how did they select them ? Please describe how they selected these states and villages.

2. Exploratory factor analysis: because the SRQ is dichotomous, factor analysis cannot be done.

I wonder if authors conducted factor analysis with categorical data? If not, my suggestion is that authors only remain the result of Chronbach’s alpha value not results of factor analysis.

Statistical analysis:

1. Some variables are categorized (Table 1) examined chi2 in order to assess the group differences.

Results

2. Because of the cross-sectional study, it is not sure if authors can use the term ‘protective factor,’ ‘Risk factor’ and ‘predictors.’

Discussion

1. 301-302 In case results are not shown, it should not be discussed because this is academic paper.

Thank you.

Reviewer #2: Multidimensional predictors of common mental disorders among Indian caregivers of 6- to 24-month-old children living in villages with women’s self-help groups

PONE-D-19-27808

Poor mental health of caregivers of young children and its impact on their own as well as children’s wellbeing are now well established including some interesting findings from low and middle income countries. However it is important to understand the protective and risk factors for poor mental health of caregivers which can inform development of culturally relevant interventions for this target population.

In introduction section, Authors mentioned consequences of CMD among caregivers in terms of loss of productivity and impact on child growth and development. Though authors talked about limited health and wellbeing budget in India and also discussed women self-help groups (SHG), however, there is no information on risk factors of CMD for caregivers of young children in low resource settings such as India. Some information on content of self-help groups would be useful. Authors also need to talk about what other support programs/models exist to support mothers or caregivers of young children in Indian context. The reference mentioned on line number 53, page number 10 for relationship between CMD among caregivers and loss of productivity is not specifically for the caregivers. This is for adults with chronic medical conditions. A very similar study was published in 2011 in Korea (Bang, K.S., Chae, S.M. and Park, S.H., 2011. Depression, health status, and parenting stress of caregivers of children in poverty. Korean Parent-Child Health Journal, 14(2), pp.55-61.)

To my understanding all the caregivers were “mothers” therefore I was wonderinng why authors used the term caregivers rather than mothers. In method section authors explained the ethical considerations such as voluntary participation and ethics approval from a review board. It would be helpful to mention use of participant information sheets in local languages and how the consent was obtained from participants with limited reading and writing skills. Not clear why the researchers obtained verbal consent? Why not written consent such as thumb print? Researchers are also now using audio recorded consents.

While describing household and environmental indicators, authors mentioned caste categories and they used term “other backward classes” (OBC) on line number 163. The term seems very stigmatizing. Can we think of an alternative name that can describe this cast category. There is no information on who approached the participants, who did assessments, training and supervision of assessors. This information is important to understand the quality of the data collected.

In discussion section authors discussed difference between farmers and housewives – being farmer would be a protective factor. This needs to be discussed further. As mixed evidence exist such as research saying no association between farming and increased mental health problems (Judd et al., 2006). On other hand evidence also exist to support high rates of suicide among farmers (Sainath. P ‘Farmers suicide rates soar above the rest’. The Hindu, May 18, 2013, Mumbai).

In limitations section on page 21, 342 – 352, authors need to mention use of a self-reporting questionnaire to assess depression rather than a structured interview is also the limitation of the study. Moreover, while talking about the components/variables that authors could not assess, they also need to talk about domestic violence, possible significant contributor of poor mental health in women and especially in mothers of young children.

Reviewer #3: The topic of this manuscript is very important. However, as currently written, the paper makes only a small contribution to the existing literature on the social determinants of mental health in India/LMIC.

Introduction – a minor point in the first sentence, but ‘stress’ is not a mental disorder. It is also not accurate to state that the topic of CMD has been neglected in high poverty contexts, there is a very large literature in India, and other LMIC, on social factors in mental health. This literature should be incorporated into the introduction so that it’s clearer what contribution the current paper is making.

More detail would also be beneficial about the ‘primary caregiver’ inclusion criteria, especially in light of the SHG context and the findings with paid work. Culturally, most mothers would say that they are the primary caregivers for their young child, even if that child is being minded by someone else during the day while she is working. Hence, what is the distinction between these women being ‘mothers of young children’ vs. ‘caregivers’?

In the Statistical Analysis section, it is not described which models adjust for all other variables vs ones that only adjust for clustering and district, and the reasoning behind this decision.

This relates to a broader issue which is that with so many variables, the analysis has a slight feeling of a fishing exercise. Additionally, many of the variables are correlated with each other and likely influence mental health symptoms through shared mechanisms. This muddles both the interpretation and impact of the findings. For example, being underweight, in the poorest wealth quintile, and experiencing food insecurity are likely very related but I don’t imagine the authors would make a strong case that ‘only food insecurity matter’s while poverty does not, for mental health. What variable is ‘statistically significant’ vs. not could very easily be an artefact of the modelling strategy. A more focused analysis related to the self-help group membership would have been more informative (and this finding does get a fair amount of space in the discussion but in the results it is only one among so many variables.)

6. PLOS authors have the option to publish the peer review history of their article (what does this mean?). If published, this will include your full peer review and any attached files.

Reviewer #1: No

Reviewer #2: No

Reviewer #3: No

---

## [Author Response · Author response to Decision Letter 0]

14 Feb 2020

Please refer to the uploaded 'Response to Reviewers' document for our responses to all comments.

---

## [Decision Letter · Decision Letter 1]

13 Mar 2020

PONE-D-19-27808R1

Multidimensional predictors of common mental disorders among Indian mothers of 6- to 24-month-old children living in disadvantaged rural villages with women’s self-help groups: a cross-sectional analysis

PLOS ONE

Dear Scott,

Thank you for submitting your manuscript to PLOS ONE.  I am sorry for the confusion that one of our reviewers made by uploading a wrong file (PONE-D review - Dec 19.docx). Please ignore that file. Please could you respond to the new comments of reviewer 1 below and the comments of reviewers 3 in the previous round? You can find reviewer 3's comments in the previous decision letter. I also attached at the end of this letter.

We would appreciate receiving your revised manuscript by Apr 27 2020 11:59PM. To enhance the reproducibility of your results, we recommend that if applicable you deposit your laboratory protocols in protocols.io, where a protocol can be assigned its own identifier (DOI) such that it can be cited independently in the future. For instructions see: http://journals.plos.org/plosone/s/submission-guidelines#loc-laboratory-protocols

We look forward to receiving your revised manuscript.

Kind regards,

Thach Duc Tran, M.Sc., Ph.D.

Academic Editor

PLOS ONE

Reviewers' comments:

Reviewer's Responses to Questions

**Comments to the Author**

1. If the authors have adequately addressed your comments raised in a previous round of review and you feel that this manuscript is now acceptable for publication, you may indicate that here to bypass the “Comments to the Author” section, enter your conflict of interest statement in the “Confidential to Editor” section, and submit your "Accept" recommendation.

Reviewer #1: All comments have been addressed

2. Is the manuscript technically sound, and do the data support the conclusions?

Reviewer #1: Yes

3. Has the statistical analysis been performed appropriately and rigorously? 

Reviewer #1: Yes

4. Have the authors made all data underlying the findings in their manuscript fully available?

Reviewer #1: Yes

5. Is the manuscript presented in an intelligible fashion and written in standard English?

Reviewer #1: Yes

6. Review Comments to the Author

Reviewer #1: Plos One：

Multidimensional predictors of common mental disorders among Indian mothers of 6-

to 24-month-old children living in disadvantaged rural villages with women’s self-help

groups: a cross-sectional analysis

Thank you for reflecting my feedback on your revision.

Here are my comments

Thank you for reflecting my feedback on your revision.

Here are my comments

1. Title

Although I understand author’s point it is important to highlight self-help group, it is too long.

My suggestion is below

Multidimensional predictors of common mental disorders among Indian mothers of 6-

to 24-month-old children infants and toddlers living in disadvantaged rural villages with women’s self-help groups: a cross-sectional analysis

Introduction. 5. Strength

Ideally, these originalities can be described in earlier part such as introduction.

Methods. 2. Exploratory factor analysis

The author’s response does not make sense. Dichotomous scale cannot be applied to (general) exploratory factor analysis statistically. My suggestion is to delete the description of exploratory factor analysis. (Line 150-153, P7)

7. PLOS authors have the option to publish the peer review history of their article (what does this mean?). If published, this will include your full peer review and any attached files.

Reviewer #1: Yes: Dr. Mizuki Takegata

REVIEWER #3 comments:

Reviewer #3: The topic of this manuscript is very important. However, as currently written, the paper makes only a small contribution to the existing literature on the social determinants of mental health in India/LMIC.

Introduction – a minor point in the first sentence, but ‘stress’ is not a mental disorder. It is also not accurate to state that the topic of CMD has been neglected in high poverty contexts, there is a very large literature in India, and other LMIC, on social factors in mental health. This literature should be incorporated into the introduction so that it’s clearer what contribution the current paper is making.

More detail would also be beneficial about the ‘primary caregiver’ inclusion criteria, especially in light of the SHG context and the findings with paid work. Culturally, most mothers would say that they are the primary caregivers for their young child, even if that child is being minded by someone else during the day while she is working. Hence, what is the distinction between these women being ‘mothers of young children’ vs. ‘caregivers’?

In the Statistical Analysis section, it is not described which models adjust for all other variables vs ones that only adjust for clustering and district, and the reasoning behind this decision.

This relates to a broader issue which is that with so many variables, the analysis has a slight feeling of a fishing exercise. Additionally, many of the variables are correlated with each other and likely influence mental health symptoms through shared mechanisms. This muddles both the interpretation and impact of the findings. For example, being underweight, in the poorest wealth quintile, and experiencing food insecurity are likely very related but I don’t imagine the authors would make a strong case that ‘only food insecurity matter’s while poverty does not, for mental health. What variable is ‘statistically significant’ vs. not could very easily be an artefact of the modelling strategy. A more focused analysis related to the self-help group membership would have been more informative (and this finding does get a fair amount of space in the discussion but in the results it is only one among so many variables.)

---

## [Author Response · Author response to Decision Letter 1]

8 Apr 2020

Please refer to the uploaded response to reviewers file and other log file.

---

## [Decision Letter · Decision Letter 2]

6 May 2020

Multidimensional predictors of common mental disorders among Indian mothers of 6- to 24-month-old children living in disadvantaged rural villages with women’s self-help groups: a cross-sectional analysis

PONE-D-19-27808R2

Dear Dr. Scott,

We are pleased to inform you that your manuscript has been judged scientifically suitable for publication and will be formally accepted for publication once it complies with all outstanding technical requirements.

With kind regards,

Thach Duc Tran, M.Sc., Ph.D.

Academic Editor

PLOS ONE

Additional Editor Comments (optional):

Reviewers' comments:

Reviewer's Responses to Questions

**Comments to the Author**

1. If the authors have adequately addressed your comments raised in a previous round of review and you feel that this manuscript is now acceptable for publication, you may indicate that here to bypass the “Comments to the Author” section, enter your conflict of interest statement in the “Confidential to Editor” section, and submit your "Accept" recommendation.

Reviewer #1: All comments have been addressed

Reviewer #3: All comments have been addressed

2. Is the manuscript technically sound, and do the data support the conclusions?

Reviewer #1: Yes

Reviewer #3: Yes

3. Has the statistical analysis been performed appropriately and rigorously? 

Reviewer #1: Yes

Reviewer #3: Yes

4. Have the authors made all data underlying the findings in their manuscript fully available?

Reviewer #1: Yes

Reviewer #3: (No Response)

5. Is the manuscript presented in an intelligible fashion and written in standard English?

Reviewer #1: Yes

Reviewer #3: (No Response)

6. Review Comments to the Author

Reviewer #1: Thank you for considering about my previous comments.

I confirmed responses and amendments of the authors.

Hopefully the manuscript will be published soon.

Reviewer #3: (No Response)

7. PLOS authors have the option to publish the peer review history of their article (what does this mean?). If published, this will include your full peer review and any attached files.

Reviewer #1: No

Reviewer #3: No

---

## [Editor Report · Acceptance letter]

12 Jun 2020

PONE-D-19-27808R2 

Multidimensional predictors of common mental disorders among Indian mothers of 6- to 24-month-old children living in disadvantaged rural villages with women’s self-help groups: a cross-sectional analysis 

Dear Dr. Scott:

I'm pleased to inform you that your manuscript has been deemed suitable for publication in PLOS ONE. Congratulations! Your manuscript is now with our production department. 

Kind regards, 

on behalf of

Dr. Thach Duc Tran 

Academic Editor

PLOS ONE